# Variant connective tissue (joint hypermobility) and its relevance to depression and anxiety in adolescents: a cohort-based case–control study

Jessica A Eccles ![ORCID],[1,2] Lisa Quadt ![ORCID],[1,2] Hannah McCarthy,[1,3] Kevin A Davies,[4] Rod Bond,[5] Anthony S David,[6] Neil A Harrison,[1,7] Hugo D Critchley[1,2]

[1]Department of Neuroscience, Brighton and Sussex Medical School, Brighton, UK
[2]Sussex Partnership NHS Foundation Trust, Worthing, UK
[3]University Hospitals Dorset NHS Foundation Trust, Poole, UK
[4]Department of Clinical and Experimental Medicine, Brighton and Sussex Medical School, Brighton, UK
[5]School of Psychology, University of Sussex, Brighton, UK
[6]Institute of Mental Health, UCL, London, UK
[7]Brain Research Imaging Centre, Cardiff University, Cardiff, UK

**Correspondence to**
Dr Jessica A Eccles;
j.eccles@bsms.ac.uk

## ABSTRACT

**Objective** To test whether variant connective tissue structure, as indicated by the presence of joint hypermobility, poses a developmental risk for mood disorders in adolescence.

**Design** Cohort-based case–control study.

**Setting** Data from the Avon Longitudinal Study of Parents and Children (ALSPAC) were interrogated.

**Participants** 6105 children of the ALSPAC cohort at age 14 years old, of whom 3803 also were assessed when aged 18 years.

**Main outcome measures** In a risk analysis, we examined the relationship between generalised joint hypermobility (GJH) at age 14 years with psychiatric symptoms at age 18 years. In an association analysis, we examined the relationship between presence of symptomatic joint hypermobility syndrome (JHS) and International Classification of Diseases-10 indication of depression and anxiety (Clinical Interview Schedule Revised (CIS-R), Anxiety Sensitivity Index) at age 18 years.

**Results** GJH was more common in females (n=856, 28%) compared with males (n=319, 11%; OR: 3.20 (95% CI: 2.78 to 3.68); p<0.001). In males, GJH at age 14 years was associated with depression at 18 years (OR: 2.10 (95% CI: 1.17 to 3.76); p=0.013). An index of basal physiological arousal, elevated resting heart rate, mediated this effect. Across genders, the diagnosis of JHS at age 18 years was associated with the presence of depressive disorder (adjusted OR: 3.53 (95% CI: 1.67 to 7.40); p=0.001), anxiety disorder (adjusted OR: 3.14 (95% CI: 1.52 to 6.46); p=0.002), level of anxiety (B=8.08, $t(3278)$=3.95; p<0.001) and degree of psychiatric symptomatology (B=5.89, $t(3442)$=5.50; p<0.001).

**Conclusions** Variant collagen, indexed by joint hypermobility, is linked to the emergence of depression and anxiety in adolescence, an effect mediated by autonomic factors in males. Recognition of this association may motivate further evaluation, screening and interventions to mitigate development of psychiatric disorders and improve health outcomes.

## INTRODUCTION

Adolescence is a critical time in psychological and physical development. An individual's trajectory from child into adult is influenced

---

**STRENGTHS AND LIMITATIONS OF THIS STUDY**

⇒ We used data from a large birth cohort study to investigate whether joint hypermobility increases the risk for depression and anxiety in young people.
⇒ Data points were used from ages 14 and 18, where hypermobility, depression and anxiety were assessed in the original birth cohort.
⇒ Available data points were limited due to the design of the original birth cohort study.

---

by interacting biological, psychological, environmental, social and cultural factors that determine the subsequent life journey. One quarter of the world's population is aged between 10 and 24 years, and around 75% of psychiatric disorders emerge during adolescence with most fully expressed before age 25 years. Psychiatric disorders in young people now represent half of the global burden of disease, and frequently carry poor long-term outcomes.[1] Longitudinal studies can provide valuable insight into the development of psychiatric disorders.[2] Furthermore, identification of risk factors can inform preventative and early interventional strategies to promote well-being and reduce future morbidity.[3 4]

Generalised joint hypermobility (GJH) is a characteristic marker of hereditary disorders of connective tissue, compromising a matrix of proteins that includes collagens, elastins, fibrillins and tenascins.[5] Joint hypermobility is typically assessed using the Beighton scale,[6] and is more common in females, declining with age. GJH itself is not necessarily a medical problem, but clinical phenotypes including hypermobile EDS (hEDS; formerly known as EDS hypermobility type/EDS type-III/ joint hypermobility syndrome; JHS),[7] and hypermobility spectrum disorder[8] (including GJH) are associated with clinically significant

issues. General population estimates show that approximately 20% of adolescents have GJH.[9]

GJH has been robustly associated with psychiatric conditions and psychological symptoms,[10] where adults with GJH are over-represented among patients expressing common mental disorders; that is, depression, anxiety and panic.[11] A nationwide population-wide cohort study demonstrates association between hypermobility and a variety of psychiatric disorders.[12] Meta-analysis of over 4000 participants suggests odds of demonstrating anxiety in adulthood are 4.39 in hypermobile compared with non-hypermobile people.[13] It is suggested that this relationship in anxiety is related to associated dysautonomia.[14 15]

Correspondingly, neuroimaging studies of individuals with joint hypermobility have identified structural and functional differences within brain regions implicated in emotional arousal, reactivity and feelings, notably amygdala[16] and insula.[17] These observational findings, however, cannot infer a causal relationship, but are of interest in this context.

Differences in cardiovascular autonomic activity (eg, heart rate) between individuals with and without psychiatric conditions have been observed, where higher resting heart rate predicts subsequent psychological symptoms in a large longitudinal cohort study.[18 19] Symptoms of depression are associated with altered autonomic activity,[20 21] such as increases in electrodermal activity,[22] skin temperature and respiratory frequency,[23] and decreased heart rate variability.[24]

Joint hypermobility is also linked to autonomic problems (dysautonomia),[25] often expressed as postural tachycardia syndrome (PoTS), a condition characterised by accelerated heart rate on standing, and other symptoms that overlap phenomenologically with anxiety.[26] Adolescents with PoTS have greater resting heart rates than those without PoTS.[27] In children and adolescents, there remains little information about how joint hypermobility might be associated with common psychiatric conditions.[28] However, a recent cross-sectional case–control study compared 93 children between the ages of 8 and 15 years with anxiety disorders with 100 age-matched and sex-matched children without anxiety disorders, and found that according to Beighton criteria, 52.7% of children with anxiety disorders had GJH.[29] In contrast, 16% of children in the comparison group had GJH, and age was found to be a risk factor, whereas sex, severity and type of anxiety disorder and attention deficit hyperactivity disorder (ADHD) were not predictive of GJH.

The objective of this present study was to test, for the first time longitudinally, whether there is an association between variant collagen, indexed by joint hypermobility (GJH and JHS) identified in early adolescence, and subsequent risk of depression and anxiety in later adolescence using a population-based birth cohort. We also sought to quantify this relationship and identify potential mediating factors.

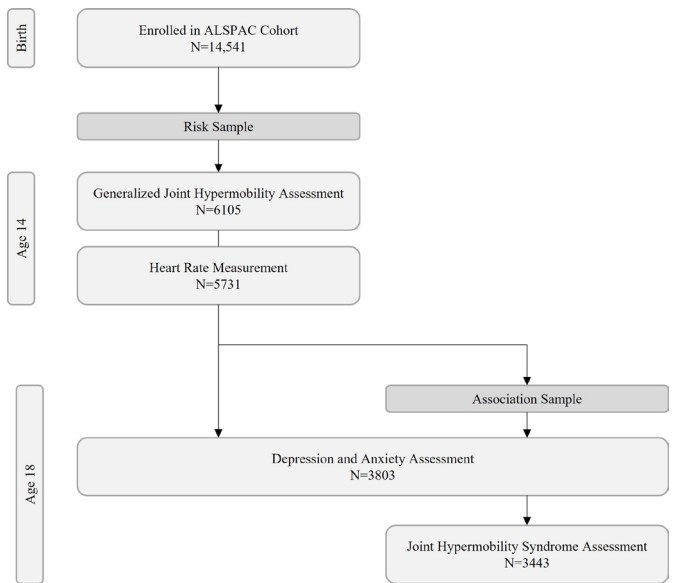

**Figure 1** Flow diagram of participants in study samples. ALSPAC, Avon Longitudinal Study of Parents and Children.

## METHODS
### Study design and participants
In this cohort-based case–control study, we used data from the The Avon Longitudinal Study of Parents and Children (ALSPAC) birth cohort to establish risk of psychiatric disorders for children and young people with and without GJH. The ALSPAC birth cohort follows children of 14 062 live births from women living in (the former) Avon County, a geographically defined region in southwest England, with expected dates of delivery between April 1991 and December 1992 (http://www.bristol.ac.uk/alspac).[30] Parents completed regular postal questionnaires about all aspects of their child's health and development from birth. From the age of 7 years, children attended annual assessment clinics, during which they underwent a variety of biomedical and physical tests, face-to-face interviews and standardised assessments of mental health. Data collection followed written informed consent, where parents of children under 16 years gave informed consent with the right to withdraw consent at any time. Additional assent from children younger than 16 years was sought at each point of data collection, where appropriate. Participants aged 16 years and older gave informed consent (given capacity to make an informed decision).[31]

Analyses for our study were based on two samples, which significantly overlap as they were from the same cohort sample: one sample was used for a predictor model, and the second for an association model (see figure 1). The 'risk set' for the predictor model included 6105 adolescents who underwent a complete joint hypermobility assessment in a clinical setting as 14 years old adolescents. Of this risk set, 5731 adolescents also had heart rate data recorded at age 14. Of these, 3803 adolescents completed assessments for depression and anxiety (Clinical Interview Schedule Revised; CIS-R)[32] aged 18 years. As per

ALSPAC dataset, these were broad time points and individual age in months varied, and therefore association tests were performed to see if age was a significant variable of interest. The second (association) sample included the 4390 adolescents who completed assessments for both joint hypermobility and the CIS-R aged 18 years. Finally, data from a subset of 3443 adolescents from the association sample, who also completed an assessment for chronic widespread pain (CWP) at the same 18-year time point, were additionally analysed. We chose the ages of 14 and 18, as these were timepoints in the ALSPAC study at which joint hypermobility was assessed before adulthood. Depression and anxiety were assessed at age 18 together with joint hypermobility, which presented an opportune second time point.

## Outcomes

ALSPAC participants underwent measurement of GJH in a dedicated research clinic, by trained assessors using the modified Beighton 9-point scoring system[6] at two time points; age 14 years and age 18 years. Each joint was assessed separately. The fifth metacarpophalangeal joint was scored as hypermobile if it could be extended >90°, the thumb if it could be opposed to the wrist, and the elbows and knees if they could be extended >10°. The spine was scored as hypermobile if both palms could be placed flat on the floor with the knees straight. Scores were recorded for individual joints, and a total score (maximum of 9) was ascertained. In common with other investigators, we used a cut-off point of ≥4 hypermobile joints as a dichotomised definition of the presence of GJH.[9] Resting heart rate was measured in the same assessment clinic setting, using a POLAR (POLAR Electro, Kempele, Finland) heart rate watch. A 'JHS' variable was derived to determine if a participant had clinically significant hypermobility at age 18 years. JHS diagnosis is typically based on the revised Brighton criteria,[33] and requires a combination of major and minor criteria, comprising GJH in combination with significant musculoskeletal and/or connective tissue symptoms. Not all minor criteria were recorded in the ALSPAC cohort at age 18 years, thus JHS was operationalised as meeting two major criteria: the presence of GJH and concomitant CWP.[34]

Depression and anxiety were measured at age 18 years, using the self-administered computerised version of the CIS-R. CIS-R is a standardised tool for measuring depression and anxiety in community samples,[32] including this cohort. The CIS-R quantifies symptoms to assign International Classification of Diseases-10 (ICD-10) indications of depression and anxiety disorders, but does not function to exclude other psychiatric conditions which may contribute to or influence symptoms of depression and anxiety.[35 36] We derived binary variables for (1) the presence or absence of an indication of major depression and (2) presence or absence of an indication of any anxiety disorder (including generalised anxiety disorder, panic disorder, agoraphobia, social phobia and specific phobia). In addition, we derived two severity indices: (3) degree of psychiatric symptomatology (ie, total CIS-R score) and (4) degree of anxiety from the Anxiety Sensitivity Index (total of 'mental and physical concerns' subscale).

## Statistical analyses

Statistical analyses were conducted in IBM SPSS Statistics V.26 and MPlus V.8.[37] We tested the hypothesis that joint hypermobility was a predictor for subsequent depression or anxiety at age 18 years, by first using logistic regression models to calculate the ORs and 95% CI for depressive disorder and/or anxiety disorder at age 18 years, separately for individuals with and without GJH at age 14 years. Since joint hypermobility is more common in females, analyses for males and females were performed separately. To explore possible associations with dysautonomia (PoTS), we conducted separate logistic regression models to test the effect of resting heart rate (as an index of physiological arousal) at age 14 years, on both GJH at age 14 years and on the later expression of depression and anxiety at age 18 years.

We then performed a mediation analysis[38] with presence of depression at age 18 years as the binary outcome variable, GJH at age 14 years as the predictor and heart rate at age 14 years as a potential mediating variable. We used probit regression with full information maximum likelihood (MPlus estimator command *ML* with Monte Carlo integration). Bootstrapping (n=2000) was applied, with the generation of 95% bias-corrected bootstrap CIs to test inferentially for direct and indirect mediation effects. Analyses were performed separately for males and females. Mediation analyses were controlled for mean-centred age at assessment and body mass index (BMI), as both were associated with the potential mediator (heart rate).

In order to test for an association between symptomatic hypermobility (JHS) and concurrent psychiatric disorders, separate logistic regression models were used to calculate ORs and 95% CI for depression and anxiety disorder at age 18 years in adolescents with and without JHS. Linear regression was used to test for differences between individuals with and without JHS in total CIS-R score and anxiety sensitivity. As data were collected at the same time point, these analyses were not subject to maximum likelihood estimation.

For our principle longitudinal analyses, we applied full information maximum likelihood (FIML) estimation,[39] where a likelihood function calculates the relationship between a probability estimate based on observed data from variables in the respective analytical model and different estimate values. FIML chooses parameter estimates that maximise this likelihood function based on complete cases, thereby providing robust parameter estimates, despite missing data.[40]

## Patient and public Involvement

This research was motivated by ongoing Patient and Public Involvement (PPI) engagement, in which concerns were regularly expressed by parents of young

**Table 1** Participant characteristics in the two study samples

| Timepoint | Evaluation | Participants with available data (n, % male) | Age (mean±SD, range) |
|---|---|---|---|
| Risk sample | | | |
| Clinic aged 14 | Generalised joint hypermobility Heart rate | 6105 (49% male) | 13.8±0.2 years, 12.5–15.2 |
| Clinic aged 18 | Depression Anxiety | 3803 (45% male) | 17.7±0.4 years, 16.3–19.3 |
| Association sample | | | |
| Clinic aged 18 | Joint hypermobility syndrome Depression Anxiety | 3557 (42% male) | 17.7±0.4 years, 16.3–19.3 |

people with joint hypermobility. PPI members often felt dismissed by medical professionals, and regularly expressed a general need to enhance awareness of joint hypermobility as a risk factor for their children. The underappreciation of its complexity and associated risks were described as barriers to effective care. Our PPI members will help with disseminating our findings among patient groups in an accessible manner.

## RESULTS
### Participants
Participant characteristics in the two samples at assessment time points are displayed in table 1, and participant flow is displayed in figure 1.

### Prevalence of hypermobility and association with age and sex assigned at birth
Of those assessed at 14 years, 1175 (19%) had GJH as reported elsewhere.[9] GJH was more common in females (n=856, 28%) compared with males (n=319, 11%; OR: 3.20 (95% CI: 2.78 to 3.68); p<0.001) and not associated with individual variation in age in months.

Of those who had assessments for joint hypermobility and CWP at 18 years, 36 (1%) had both GJH and CWP, and thus met operationalised criteria for JHS. This was significantly more common in females; only three were males (OR: 8.04 (95% CI: 2.46 to 26.25); p=0.001). JHS was not associated with variation in age.

### Risk set analysis
#### Risk of depression and anxiety at age 18 years according to hypermobility status at age 14 years
Table 2 displays the proportion of participants with GJH at age 14 years in whom depression or anxiety was indexed at age 18 years. Across the whole group, GJH was significantly associated with depression at 18 (OR: 1.70 (95% CI: 1.31 to 2.22); p=0.001), where a greater proportion of individuals with depression were

**Table 2** Relationship between hypermobility at age 14 years and depression and anxiety aged 18 years

| | Whole group | Males | Females |
|---|---|---|---|
| No ICD-10 depression | | | |
| Patients with available data (N) | 3510 | 1627 | 1883 |
| Proportion of GJH (n, %, 95% CI) | 688 (19.6%, 18.31 to 20.96) | 166 (10.2%, 8.82 to 11.77) | 522 (27.74%, 25.76 to 29.81) |
| ICD-10 depression | | | |
| Patients with available data (N) | 293 | 78 | 215 |
| Proportion of GJH (n, %, 95% CI) | 86 (29.35%, 24.42 to 34.48) | 15 (19.23%, 12.01 to 29.33) | 71 (33.02%, 27.08 to 39.56) |
| No ICD-10 anxiety | | | |
| Patients with available data (N) | 3437 | 1604 | 1832 |
| Proportion of GJH (n, %, 95% CI) | 691 (20.10%, 18.79 to 21.47) | 168 (10.47%, 9.06 to 12.06) | 523 (28.55%, 26.53 to 30.66) |
| ICD-10 anxiety | | | |
| Patients with available data (N) | 366 | 101 | 265 |
| Proportion of GJH (n, %, 95% CI) | 83 (22.68%, 16.69 to 27.24) | 13 (12.87%, 7.68 to 20.78) | 70 (26.41%, 21.48 to 32.04) |

GJH, generalised joint hypermobility; ICD-10, International Classification of Diseases-10.

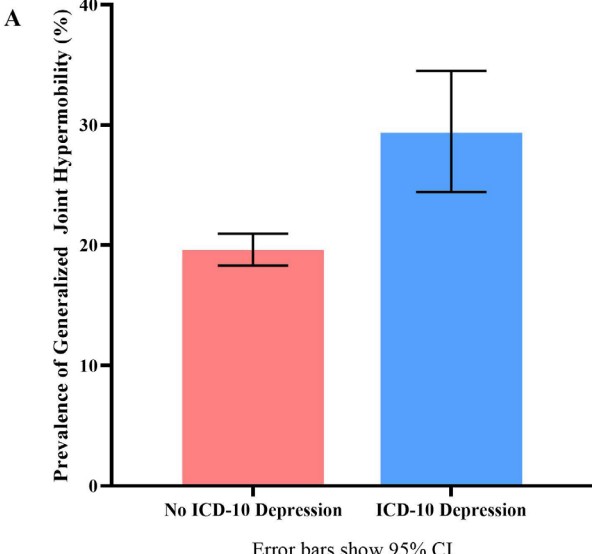

A

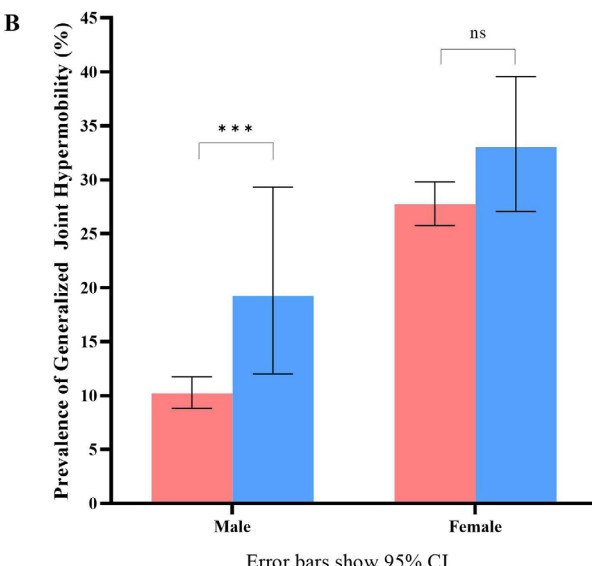

B

**Figure 2** Prevalence of generalised joint hypermobility. (A) Shows the difference in prevalence of GJH in participants who meet diagnostic criteria for ICD-10 depression (blue bar) and those who do not (red bar). (B) Shows a significant difference in prevalence of generalised joint hypermobility in depressed and non-depressed participants in male but not female participants. Error bars show 95% CI. ICD-10, International Classification of Diseases-10.

hypermobile (had GJH) than those without depression (table 2, figure 2A). Additionally, we observed a significant interaction of GJH and sex on depression ($F_{(2,3802)}$=3.73, p=0.024), where this effect was significant only in males (OR: 2.10 (95% CI: 1.17 to 3.76); p=0.013; table 2, figure 2B).

In this analysis, GJH at 14 years was not associated with anxiety at 18 years.

### Associations with heart rate at age 14 years

Heart rate was significantly higher in females (M=83.92, SD=11.94) than males (M=78.6, SD=11.44; $t(5725)$=17.18, p<0.001), and was significantly associated with variations

in exact age (months) ($r(5729)$=−0.32, p=0.016) and BMI ($r(5722)$=0.081, p<0.001) at assessment. After correcting for age and BMI, heart rate at 14 years was significantly associated with both GJH at age 14 years (OR: 1.01 (95% CI: 1.01 to 1.02); p<0.001), and also with depression at 18 years (OR: 1.01 (95% CI: 1.00 to 1.02), p=0.007).

### Risk of depression at age 18 years

The OR of depression at 18 years in both males and females, given GJH at 14 years, was 1.28 ($b$=0.246 (95% CI: 1.12 to 1.47); p<0.001). In males, the OR was 1.38 ($b$=0.032 (95% CI: 0.97 to 1.79); p=0.036). There was no significant effect in females only.

### Mediation analysis

A mediation analysis was only conducted in males, for whom the relationship between GJH and depression was significant. Both GJH and heart rate (at age 14 years) were significantly associated with depression (at age 18 years). The mediation analysis showed that the relationship between GJH and depression was mediated by heart rate (figure 3). This was demonstrated by the bootstrap CI derived from 2000 samples of the indirect effect of heart rate on the relationship between GJH and depression being significant ($b$=1.05 (95% CI: 1.011 to 1.125)) when controlling for mean centred age and BMI.

### Association set analysis

#### Symptomatic hypermobility (JHS) and depression at age 18 years

A higher proportion of JHS was found in participants who met ICD-10 criteria for depression (figure 4A). Across the whole group, JHS was significantly associated with depression at age 18 years (figure 5 shows ORs for this effect, which remained significant when adjusted for sex). Given the low levels of JHS in males (n=3), this analysis was not powered to be undertaken separately but remained significant for females (figure 5).

Across the whole group, higher CIS-R scores were present in the JHS group, compared with those without JHS (B=5.89, $t(3442)$=5.50, p<0.001; figure 6A). In analyses corrected for sex, this association remained (B=4.97, $t(3441)$=4.73, p<0.001). In females, higher CIS-R scores were present in those with JHS (M=11.97, SD=10.13), compared with those without JHS (M=6.61, SD=6.88) (B=5.36, $t(1992)$=4.40, p<0.001).

#### Symptomatic hypermobility (JHS) and anxiety at age 18

The prevalence of JHS was greater among participants who met ICD-10 criteria for an anxiety disorder (figure 4B). JHS was significantly associated with any anxiety disorder at age 18 years, a relationship that remained significant when adjusted for sex. ORs are displayed in figure 5.

Across the whole group, an elevated anxiety sensitivity score was observed in individuals with JHS, compared with those without JHS (B=8.08, $t(3278)$=3.95, p<0.001; figure 6B). This association remained after adjusting for sex (B=6.60, $t(3278)$=3.28, p=0.001). In females, higher anxiety sensitivity was present in those with JHS (M=55.94,

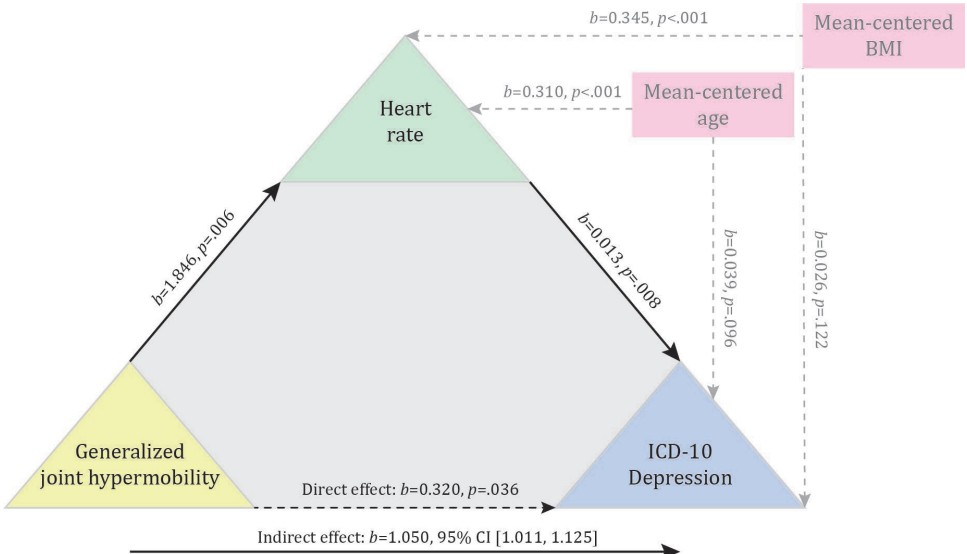

**Figure 3** Heart rate mediates the relationship between GJH and depression. This figure displays standardised regression coefficients for the relationship between generalised joint hypermobility and depression as mediated by heart rate (solid black arrows), with the direct (dashed black arrow) and significant indirect effects (solid black arrow). Effects of covariates on heart rate and depression are displayed with grey dashed arrows. BMI, body mass index.

SD=12.26), compared with those without JHS (M=49.33, SD=11.76; B=−6.61, t(1907)=−3.10, p=0.002).

## DISCUSSION

In adults, the link between variant connective tissue structure (indexed by joint hypermobility) and affective psychopathology is largely established[13] and, interestingly, reinforced by convergent evidence from domestic dogs.[41] Our results extend these findings by showing a link between GJH at age 14 years and subsequent depression in adolescence. There was a significant interaction of sex with significant effects in males only. We also show a mediating effect of basal autonomic physiology (indexed by heart rate) on this relationship. However, we found no longitudinal association between GJH alone and anxiety. This observation is striking as such an association is widely reported in adults.[10 13 42] However, symptomatic joint hypermobility (JHS) operationalised here as the presence of GJH and CWP was associated with both anxiety and depression in female adolescents and across the whole group when adjusted for sex.

Variant connective tissue, expressed as joint hypermobility, is characteristically a familial trait. However, no single gene is implicated with consistency. Nevertheless, symptomatic hypermobility (hEDS /HSD/JHS) is considered to be an autosomal dominant trait with incomplete penetrance, variable expressivity and influenced by sex, since joint hypermobility is more common in females.[43] However, despite this female preponderance, joint hypermobility is also associated with neurodivergence[44] (including Fragile X syndrome, ADHD and autism), which is more commonly diagnosed in males. Importantly, however reported co-occurrences are potentially skewed by underdiagnosis of neurodivergence in

females.[45] Neurodivergence also increases the likelihood of clinical anxiety and depression,[46] and it is plausible that this mechanism may account for the observed association between GJH at 14 years and high rates of depression at age 18 years in males.

Hypermobility is associated with dysautonomia,[47] typically PoTS, symptoms of which have phenomenological overlap with anxiety.[26] Adolescents with PoTS have greater resting heart rates than peers without PoTS.[27] Our present finding of a mediating role of heart rate on the association between hypermobility and depression may thus indicate the presence of dysautonomia and constitutional differences in physiological reactivity. Interestingly, a Swedish cohort study also demonstrated that adolescent males with higher basal heart rate are more likely to develop subsequent psychopathology, notably anxiety disorders.[19] However, it should be noted that the size of effect linking heart rate with hypermobility and depression was small.

Our data suggest that depression is associated, in males, with the presence of GJH alone. However, in females, additional features of symptomatic hypermobility (JHS), notably CWP, are required for this association to be significant. One possible explanation for this pattern of results is that the higher rates of anxiety and depression in females reflect susceptibility and perhaps exposure to a wider range of risk factors, which obscures an association with GJH apparent in males.

This study was limited by the measurement instruments used in the ALSPAC cohort study. The categorisation and classification of joint hypermobility and symptomatic joint hypermobility remains the subject of substantial debate.[8] Although the Beighton Scale remains the most widely used approach for measuring hypermobility, there

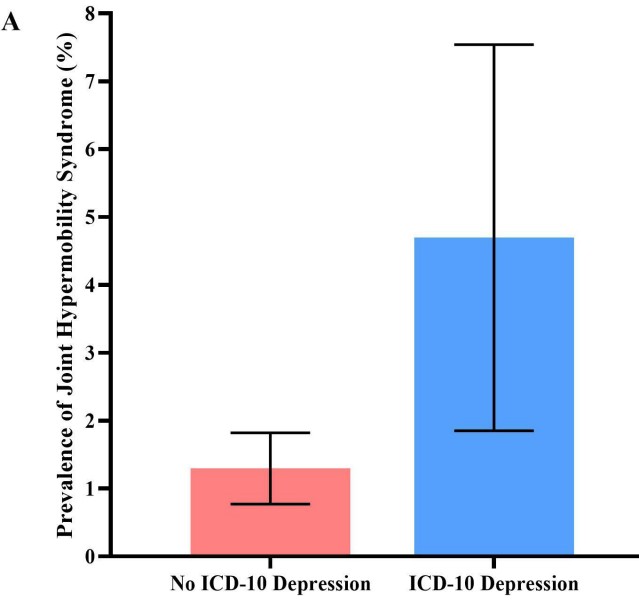

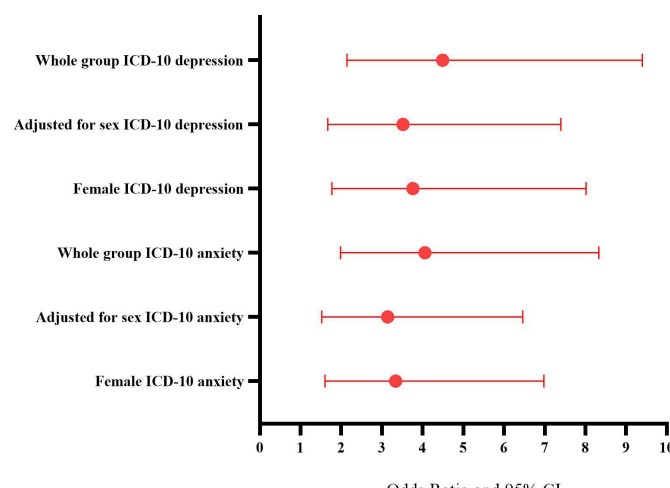

**Figure 5** Forest plot of ORs. This figure displays the ORs and their 95% CIs of depression and anxiety given the presence of joint hypermobility syndrome in the whole group, adjusted for sex, and in females only. ICD-10, International Classification of Diseases-10.

neurodevelopmental conditions like autism and ADHD, however, these are based on parent reports at age 9 years of the child and were outside of the scope of this work.

Our study provides evidence for both a longitudinal association between a common variant of connective tissue, hypermobility and a common psychiatric illness, depression, at age 18 years. We also observed more direct association between symptomatic joint hypermobility (JHS) and depression and anxiety at age 18 years. A more fine-grained assessment of mental health symptoms might have uncovered a broader range of outcomes. Recent years have seen increasing attention in the lay media to joint hypermobility as a single biomedical cause of sometimes profound disability. However, the contribution of underlying variant connective tissue structure is likely to be complex and involves interacting autonomic, psychological and physiological factors. Further systematic scientific inquiry is required.

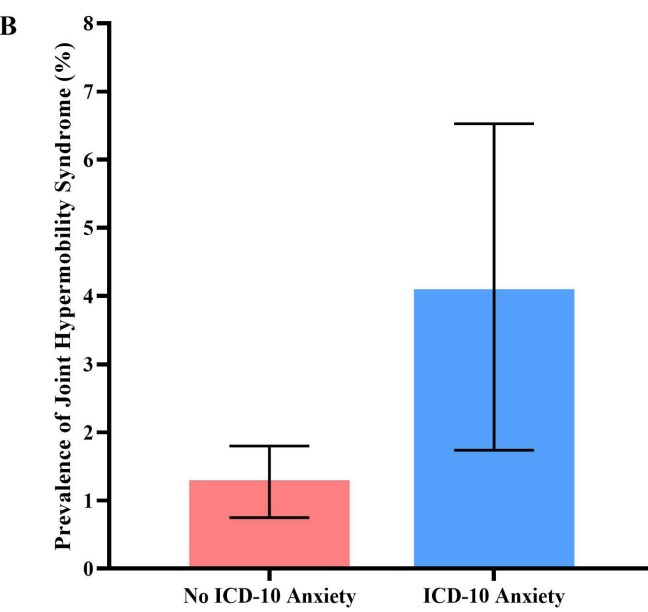

**Figure 4** Prevalence of joint hypermobility syndrome. This figure shows the difference in prevalence of joint hypermobility syndrome in participants who meet ICD-10 criteria (blue bars) for depression (A) and anxiety (B), and those who do not meet diagnostic criteria (red bars). ICD-10, International Classification of Diseases-10.

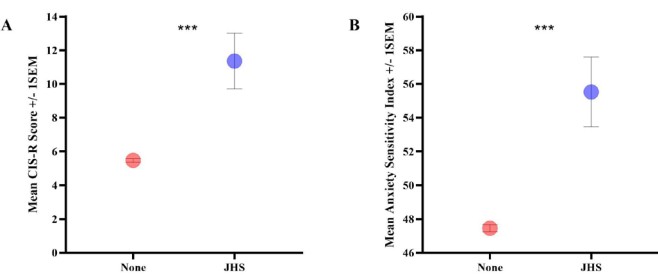

**Figure 6** Mean CIS-R scores and Anxiety Sensitivity Index scores. (A) Displays the group difference in mean CIS-R (depression) score between participants with (blue dot) and without (red dot) JHS. (B) Displays the group difference in mean Anxiety Sensitivity Index score between participants with (blue dot) and without (red dot) joint hypermobility syndrome. Error bars shows ±1 SEM. CIS-R, Clinical Interview Schedule Revised.

remains considerable questions about its utility, particularly in children and adolescents,[48] and cut-offs vary widely. While the CIS-R is a well-established tool to find robust indications of depression and anxiety, it does not control or test for other psychiatric conditions that may influence depressive and anxiety symptomatology. It is therefore only an approximation for an indication of the presence of depression and anxiety, although a tried and tested one. The ALSPAC dataset includes variables for

Together, our findings highlight the importance of a broad biopsychosocial approach to these complex conditions, highlighting the interplay between physical and mental health, with the possibility of screening for hypermobility in adolescence, the judicious use of which may provide opportunities for early intervention to identify and alleviate common psychiatric symptoms.

**Acknowledgements** We are extremely grateful to all the families who took part in this study, the midwives for their help in recruiting them and the whole ALSPAC team, which includes interviewers, computer and laboratory technicians, clerical workers, research scientists, volunteers, managers, receptionists and nurses.

**Contributors** JE acquired funding, conceptualised and designed the study with support from HDC and KAD, analysed and managed the data and drafted the initial manuscript. LQ visualised data and revised the manuscript. RB, HM and LQ supported data analysis. JE, LQ, HM, ASD, RB, NAH, KAD critically reviewed the manuscript, contributed to drafts and approved the final version. We attest all authors meet ICMJE requirements for authorship. JE acts as guarantor.

**Funding** Funding for this project came via a fellowship to JAE (MRC MR/K002643/1). JAE was also supported by MQ Transforming Mental Health and Versus Arthritis (MQF 17/19).

**Competing interests** None declared.

**Patient and public involvement** Patients and/or the public were involved in the design, or conduct, or reporting or dissemination plans of this research. Refer to the Methods section for further details.

**Patient consent for publication** Not applicable.

**Ethics approval** Approval for the original ALSPAC study was obtained from the ALSPAC Ethics and Law Committee (IRB00003312) and Local Research Ethics Committees. Participants gave informed consent to participate in the study before taking part.

**Provenance and peer review** Not commissioned; externally peer reviewed.

**Data availability statement** Data may be obtained from a third party and are not publicly available. All data are available on the ALSPAC data dictionary and variable search tool (http://www.bristol.ac.uk/alspac/researchers/our-data/).

**ORCID iDs**
Jessica A Eccles http://orcid.org/0000-0002-0062-1216
Lisa Quadt http://orcid.org/0000-0002-5896-916X

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
