## [Reviewer comments · BMJ Open]

ARTICLE DETAILS

TITLE (PROVISIONAL)	Variant connective tissue (joint hypermobility) and its relevance to depression and anxiety in adolescents: a cohort-based case-control study
AUTHORS	Eccles, Jessica; Quadt, Lisa; McCarthy, Hannah; Davies, Kevin; Bond, Rod; David, Anthony; Harrison, Neil; Critchley, Hugo

VERSION 1 – REVIEW

REVIEWER	Ishiguro, Hiroki University of Yamanashi
REVIEW RETURNED	15-Aug-2022

GENERAL COMMENTS	Eccles et al. reported the importance of additional surveillance and support system for adolescents with general joint hypermobility regarding possible risk for depressiveness and anxiety in the article “Variant connective tissue (joint hypermobility) and its relevance to depression and anxiety in adolescents: a cohort-based case-control study”. GJH includes congenital connective tissue diseases, such as Ehlers-Danlos syndrome. Many psychiatrists are less interested in physical conditions, including congenital diseases, and have not been able to adequately address the development of depressive and anxiety disorders in the AYA generation. Pediatricians, on the other hand, are unfamiliar with preventing the onset of psychiatric symptoms in the AYA generation. Thus, reporting the importance of screening for JH in adolescence regarding interactions between physical and mental health, this study provides an important rationale and a suggestion for early intervention against the risk of developing mental illness. Although the research is interesting, I have some concerns that must be addressed before the article is open to the journal audience. Major concerns: 1. In section “strength and limitation of this study”, more explanation needed why the authors choose 18 years of age that is not probable age for depression in this study.2. Please clearly explain from whom informed consents were obtained (they must be from research subjects themselves, if they were legally adult). While informed consents were obtained from the parents when they join in ALSPAC, we need to obtain informed ascent if the research subjects were not legally adult.(depending on the countries’ law).3. Regarding the participants, risk-set (first sample) and association sample (second sample) are independent? No overlapped samples used, did it? Please clarify.
---

	4. In materials and methods section, page 6, it is not clear how to make diagnosis of depression and anxiety disorders using CIS-R. Did this research exclude any other psychiatric diseases, such as schizophrenia, bipolar disorder, personality disorder. In discussion, the author discusses gender differences in neurodevelopmental disorders could affect the result of the study regarding an association between JH and psychiatric symptoms. The discussion was also unclear whether there were diagnosis for neurodevelopmental disorders in the research subjects used in the study. Please explain in detail. Minor concerns  1. In abstract page, there are missing P value in the result section. 2. In introduction page 4 line 14-15, reference is needed. 3. Page 4, please add some explanation regarding JH/EDS and psychiatric illness with references. 4. Neuro-Image is introduced in Introduction page 4. Please clarify whether those findings be phenomena cause of or the result from JH. 5. In methods page 5, "Of this risk set, 5731 adolescents also had heart rate data recorded at the same time-point." Please make it clear for "same time-point". 6. Abbreviation CWP must be used for the second and later used. 7. Please give company detail for POLAR watch, production company and place. 8. Page 8, in "Prevalence of hypermobility and association with age and sex assigned at birth" section. The sentences "not associated with variation in age" and for "JHS was not associated with variation in age" are not clear. 9. Page 8 "Additionally, we observed a significant interaction of JH". JH is GJH or JHS? 10. In Discussion section page 10, the result was missing after "(M=49.33, SD=11.76)". 11. In Discussion section page 11, "Our data suggest that depression can be predicted, in males, by the presence of GJH." The sentence sounds too deterministic. 12. Data Sharing section, "." was missing.
--	---

REVIEWER	Zahed , Ghazal Shahid Beheshti University of Medical Sciences, Child and adolescent psychiatry
REVIEW RETURNED	22-Aug-2022

GENERAL COMMENTS	Thank you for the opportunity to review the manuscript titled, "Variant connective tissue (joint hypermobility) and its relevance to depression and anxiety in adolescents: a cohort-based case-control study" for the BMJ. Overall, the article was well-written, However, I have several concerns about the manuscript in its current state.  1. In many parts of the manuscript, it was mentioned that GJH is predictor of depression or is strongly related to it. Considering that there is still no certainty about the cause-and-effect relationship between these two cases, it is better to use expressions such as comorbidity or coexistence. (e.g. Page 3, lines 29-30; page 9, lines 48-49). 2. In the abstract, it was mentioned that the diagnosis of JHS was associated with the presence of depression and anxiety disorder. Considering that you used the self-administered computerized
---

	version of the Clinical Interview Schedule-Revised (CIS-R) to assess anxiety and depression, you cannot use the word “disorder” and you can only describe the symptoms of depression and anxiety. 3. On page 5, the last two lines, you mentioned the objective of this study was to test, for the first time longitudinally, whether there is an association between variant collagen, indexed by joint hypermobility (GJH and JHS) identified in early adolescence, and subsequent risk of depression and anxiety in later adolescence using a population-based birth cohort. A research in 2020 similar to yours was conducted by Dr. Javadi Parvaneh et al. Please state the reasons why your research is different from Dr. Javadi's research. 4. You mentioned that after correcting for age and BMI, heart rate at 14 years was significantly associated with both GJH at age 14 years, and also with depression at 18 years (page 10, lines 33-36), elevated resting heart rate, mediated this effect. (Abstract, line 32). Please add more information about physiological changes in depression. Thank you,
--	---

VERSION 1 – AUTHOR RESPONSE

Reviewer: 1

Dr. Hiroki Ishiguro, University of Yamanashi

Comments to the Author:

Eccles et al. reported the importance of additional surveillance and support system for adolescents with general joint hypermobility regarding possible risk for depressiveness and anxiety in the article “Variant connective tissue (joint hypermobility) and its relevance to depression and anxiety in adolescents: a cohort-based case-control study”. GJH includes congenital connective tissue diseases, such as Ehlers-Danlos syndrome. Many psychiatrists are less interested in physical conditions, including congenital diseases, and have not been able to adequately address the development of depressive and anxiety disorders in the AYA generation. Pediatricians, on the other hand, are unfamiliar with preventing the onset of psychiatric symptoms in the AYA generation. Thus, reporting the importance of screening for JH in adolescence regarding interactions between physical and mental health, this study provides an important rationale and a suggestion for early intervention against the risk of developing mental illness.

Although the research is interesting, I have some concerns that must be addressed before the article is open to the journal audience.

We thank the reviewer for their encouraging assessment of our work and have addressed their concerns below:

Major concerns:

1. In section “strength and limitation of this study”, more explanation needed why the authors choose 18 years of age that is not probable age for depression in this study.

We understand that the ages in the study may seem arbitrary and have added an explanation in the Strengths and Limitations and Methods section in the paper. The age was chosen as this was the only time point at which depression was recorded using self-report rather than parent-report in the ALSPAC study. We used this openly accessible data base as it presents a large birth cohort with variables of significant interest. However, we cannot influence at which

point certain variables were collected and have therefore chosen the ages of 14 and 18, as both were the only ages at which joint hypermobility was assessed.

Strengths and Limitations:

- We used data from a large birth-cohort study to investigate whether joint hypermobility increases the risk for depression and anxiety in young people
- Data points were used from ages 14 and 18, where hypermobility, depression, and anxiety were assessed in the original birth-cohort.
- Available data points were limited due to the design of the original birth-cohort study.
- We used a birth-cohort in a longitudinal case-control study to investigate if hypermobility at age 14 was a predictor for psychiatric symptoms at age 18 in males and females.
- In a risk analysis, we tested for a relationship between Generalized Joint Hypermobility at age 14 years and depression and anxiety at age 18, and whether physiological arousal had a mediating role on this relationship.
- In an association analysis, we examined the effects of symptomatic Joint Hypermobility Syndrome on psychiatric symptoms in males and females at 18 years.
- Classification of (symptomatic) joint hypermobility is still heavily debated, with varying cut-offs and remaining questions about the Beighton Scale as the most widely used measurement tool.
- Contributions of variant connective tissue/joint hypermobility are likely complex and will require more systematic scientific inquiry.

Methods:

We chose the ages of 14 and 18, as these were the only timepoints in the ALSPAC study at which joint hypermobility was assessed. Depression and anxiety were assessed at age 18 together with joint hypermobility, which presented an opportune second time point. (p 6, lines 32-38)

2. Please clearly explain from whom informed consents were obtained (they must be from research subjects themselves, if they were legally adult). While informed consents were obtained from the parents when they join in ALSPAC, we need to obtain informed ascent if the research subjects were not legally adult.(depending on the countries' law).

We have clarified the consent procedures from the ALSPAC study:

Data collection followed written informed consent, where parents of children under 16 years gave informed consent with the right to withdraw consent at any time. Additional assent from children younger than 16 years was sought at each point of data collection were appropriate. Participants aged 16 years and older gave informed consent (given capacity to make an informed decision.¹⁹ (p 6, lines 3-10)

3. Regarding the participants, risk-set (first sample) and association sample (second sample) are independent? No overlapped samples used, did it? Please clarify.

The samples are not independent, as they are from the same cohort. However, as not all participants took part in all aspects of each time point of data collection, the numbers differ, while the sample does overlap. We have clarified this in the Methods section:

Analyses for our study were based on two samples, which significantly overlap as they were from the same cohort sample:[...] (p 6, lines 12-14)

4. In materials and methods section, page 6, it is not clear how to make diagnosis of depression and anxiety disorders using CIS-R. Did this research exclude any other psychiatric diseases, such as schizophrenia, bipolar disorder, personality disorder. In discussion, the author discusses gender differences in neurodevelopmental disorders could affect the result of the study regarding an association between JH and psychiatric symptoms. The discussion was also unclear whether there were diagnosis for neurodevelopmental disorders in the research subjects used in the study. Please explain in detail.

We have addressed this shortcoming in the methods and discussion sections to clarify the scope of the CIS-R and this study:

Methods:

The CIS-R quantifies symptoms to assign International Classification of Diseases-10 (ICD-10) indications of depression and anxiety disorders, but does not function to exclude other psychiatric conditions which may contribute to or influence symptoms of depression and anxiety.^{27,28} We derived binary variables for; 1) the presence or absence of an indication of major depression, and; 2) presence or absence of an indication of any anxiety disorder (including generalized anxiety disorder, panic disorder, agoraphobia, social phobia and specific phobia). (p 7, lines 19-29)

Discussion:

This study was limited by the measurement instruments used in the ALSPAC cohort study. The categorization and classification of joint hypermobility and symptomatic joint hypermobility remains the subject of substantial debate.⁸ Although the Beighton Scale remains the most widely-used approach for measuring hypermobility, there remain considerable questions about its utility, particularly in children and adolescents,⁴¹ and cut-offs vary widely. While the CIS-R is a well-established tool to find robust indications of depression and anxiety, it does not control or test for other psychiatric conditions that may influence depressive and anxiety symptomatology. It therefore only an approximation for an indication of the presence of depression and anxiety, albeit a tried and tested one. The ALSPAC data set includes variables for neurodevelopmental conditions like Autism and ADHD, however, these are based on parent reports at age 9 years of the child and were outside of the scope of this work. (p 12, lines 53-60 & p 13, lines 3-14)

Minor concerns

1. In abstract page, there are missing P value in the result section.

We have corrected this:

In males only, GJH at age 14 years predicted depression at 18 years (OR 2.10 (95%CI 1.17 to 3.76); P=0.013) an index of basal physiological arousal, elevated resting heart rate, mediated this effect. Across genders, the diagnosis of JHS at age 18 years was associated with the presence of depressive disorder (adjusted OR 3.53 (95%CI 1.67 to 7.40); P=0.001), anxiety disorder (adjusted OR 3.14 (95% CI 1.52 to 6.46); P=0.002), level of anxiety (B=8.08, t(3278)=3.95; P<0.001), and degree of psychiatric symptomatology (B=5.89, t(3441)=0.093; p<0.001). (p 2, , lines 29-41)

2. In introduction page 4 line 14-15, reference is needed.

We have added the following reference: Fusar-Poli P. Integrated Mental Health Services for the Developmental Period (0 to 25 Years): A Critical Review of the Evidence. *Front Psychiatry* 2019;10:355. doi: 10.3389/fpsy.2019.00355 [published Online First: 2019/06/25]

3. Page 4, please add some explanation regarding JH/EDS and psychiatric illness with references.

We have added the following explanations with references:

GJH has been robustly associated with psychiatric conditions and psychological symptoms,¹⁰ where adults with GJH are over-represented among patients expressing common mental disorders; i.e. depression, anxiety and panic.¹¹ A nationwide population-wide cohort study demonstrates association between hypermobility and a variety of psychiatric disorders.¹² Meta-analysis of over 4000 participants suggests odds of demonstrating anxiety in adulthood are 4.39 in hypermobile compared to non-hypermobile people.¹³ It is suggested that this relationship in anxiety is related to associated dysautonomia.^{14,15} (p 4, lines 39-50)

4. Neuro-Image is introduced in Introduction page 4. Please clarify whether those findings be phenomena cause of or the result from JH.

We have added this clarification:

Correspondingly, neuroimaging studies of individuals with joint hypermobility have identified structural and functional differences within brain regions implicated in emotional arousal, reactivity and feelings, notably amygdala¹⁶ and insula.¹⁷ These observational findings, however, cannot infer a causal relationship, but are of interest in this context. (p 4, lines 55-57)

5. In methods page 5, "Of this risk set, 5731 adolescents also had heart rate data recorded at the same time-point." Please make it clear for "same time-point".

Of this risk set, 5731 adolescents also had heart rate data recorded at age 14 the same time-point. (p 6, line 20)

6. Abbreviation CWP must be used for the second and later used.

We have changed these instances:

Finally, data from a subset of 3443 adolescents from the association sample, who also completed an assessment for chronic widespread pain (CWP) at the same 18-year time point,[...] (p 6, line 33)

Not all minor criteria were recorded in the ALSPAC cohort at age 18 years, thus JHS was operationalized as meeting two major criteria: the presence of GJH and concomitant chronic widespread pain (CWP).²² (p 6, line 12)

Of those who had assessments for joint hypermobility and chronic widespread pain (CWP) at 18 years, 36 (1%) had both GJH and CWP, and thus met operationalized criteria for JHS. (p 9, line 32)

However, symptomatic joint hypermobility (JHS; operationalized here as the presence of GJH and CWP chronic widespread pain) predicted both anxiety and depression in female adolescents and across the whole group when adjusted for sex. (p 12, line 3)

However, in females, additional features of symptomatic hypermobility (JHS), notably chronic widespread pain CWP, are required. (p 12, line 45)

7. Please give company detail for POLAR watch, production company and place.

We have added the following details:

Resting heart rate was measured in the same assessment clinic setting, using a POLAR (POLAR Electro, Kempele, Finland) heart rate watch. (p 6, lines 58-60)

8. Page 8, in "Prevalence of hypermobility and association with age and sex assigned at birth" section. The sentences "not associated with variation in age" and for "JHS was not associated with variation in age" are not clear.

We have clarified this in the methods and results sections:

Methods:

Of these, 3803 adolescents completed assessments for depression and anxiety (Clinical Interview Schedule Revised; CIS-R)²⁴ aged 18 years. As per ALSPAC data set, these were broad time points and individual age in months varied, and therefore association tests were performed to see if age was a significant variable of interest. (p 6, lines 22-26)

Results:

GJH was more common in females (n=856, 28%) compared to males (n=319, 11%; OR 3.20 [95% CI 2.78-3.68]; p<0.001) and not associated with individual variation in age in months. (p9, line 30)

9. Page 8 "Additionally, we observed a significant interaction of JH". JH is GJH or JHS?

We have corrected this typo:

Additionally, we observed a significant interaction of GJH and sex on depression[...] (p 9, line 53)

10. In Discussion section page 10, the result was missing after "(M=49.33, SD=11.76)".

We have added the result:

In females, higher anxiety sensitivity was present in those with JHS (M=55.94, SD=12.26), compared to those without JHS (M=49.33, SD=11.76; B=-6.61, t(1907)=-3.10, p=0.002). (p 11, line 41)

11. In Discussion section page 11, "Our data suggest that depression can be predicted, in males, by the presence of GJH." The sentence sounds too deterministic.

We have revised this sentence:

Our data suggest that depression is associated, in males, with the presence of GJH alone. However, in females, additional features of symptomatic hypermobility (JHS), notably chronic widespread pain CWP, are required for this association to be significant. (p 12, lines 43-47)

12. Data Sharing section, "." was missing.

We have corrected this typo:

All data is available on the ALSPAC data dictionary and variable search tool. (p 14, line 39)

Reviewer: 2

Dr. Ghazal Zahed , Shahid Beheshti University of Medical Sciences

Comments to the Author:

Thank you for the opportunity to review the manuscript titled, "Variant connective tissue (joint hypermobility) and its relevance to depression and anxiety in adolescents: a cohort-based case-control study" for the BMJ. Overall, the article was well-written, However, I have several concerns about the manuscript in its current state.

We thank the reviewer for their positive appraisal of our manuscript. We have addressed their concerns as follows:

1. In many parts of the manuscript, it was mentioned that GJH is predictor of depression or is strongly related to it. Considering that there is still no certainty about the cause-and-effect relationship between these two cases, it is better to use expressions such as comorbidity or coexistence. (e.g. Page 3, lines 29-30; page 9, lines 48-49).

We have toned down how we frame the detected relationship between GJH and depression/anxiety throughout the manuscript. However, where we describe a statistical relationship (i.e., predictors in regression models), we kept the correct terminology.

Abstract:

In males only, GJH at age 14 years was associated with depression at 18 years (OR 2.10 (95%CI 1.17 to 3.76); P=0.013). (p 2, lines 30-32)

Results:

Across the whole group, GJH was significantly associated with depression at 18 (OR 1.70 [95% CI 1.31-2.22]; p=0.001), where a greater proportion of individuals with depression were hypermobile (had GJH) than those without depression (Table 2, Figure 2A). (p 9, lines 48-50)

Discussion:

However, symptomatic joint hypermobility (JHS; operationalized here as the presence of GJH and CWP chronic widespread pain) was associated with both anxiety and depression in female adolescents and across the whole group when adjusted for sex. (p 11, line 60 & p 12 lines 3-4) Our data suggest that depression is associated, in males, with the presence of GJH alone. However, in females, additional features of symptomatic hypermobility (JHS), notably chronic widespread pain CWP, are required for this association to be significant. (p 12, lines 43-46)

2. In the abstract, it was mentioned that the diagnosis of JHS was associated with the presence of depression and anxiety disorder. Considering that you used the self-administered computerized version of the Clinical Interview Schedule-Revised (CIS-R) to assess anxiety and depression, you cannot use the word "disorder" and you can only describe the symptoms of depression and anxiety.

We have revised this throughout the abstract and methods:

Abstract

In an association analysis, we examined the relationship between presence of symptomatic joint hypermobility syndrome (JHS) and International Classification of Diseases (ICD-10) indication of depression and anxiety (Clinical Interview Schedule Revised [CIS-R], Anxiety Sensitivity Index [ASI]) at age 18 years. (p 2, lines 23-27)

Methods

The CIS-R quantifies symptoms to assign International Classification of Diseases-10 (ICD-10) indications of depression and anxiety disorders.^{23,24} We derived binary variables for; 1) the presence or absence of an indication of major depression, and; 2) presence or absence of an indication of any anxiety disorder (including generalized anxiety disorder, panic disorder, agoraphobia, social phobia anspecific phobia). (p 7, lines 19-29)

3. On page 5, the last two lines, you mentioned the objective of this study was to test, for the first time longitudinally, whether there is an association between variant collagen, indexed by joint hypermobility (GJH and JHS) identified in early adolescence, and subsequent risk of depression and anxiety in later adolescence using a population-based birth cohort. A

research in 2020 similar to yours was conducted by Dr. Javadi Parvaneh et al. Please state the reasons why your research is different from Dr. Javadi's research.

We have added Dr Javadi's research to the introduction, showing that their study was a cross-sectional and smaller study:

Nevertheless, In children and adolescents, there remains little information about how joint hypermobility might be associated with common psychiatric conditions.²¹ However, a recent cross-sectional case-control study compared 93 children between the ages of 8-15 years with anxiety disorders with 100 age and sex-matched children without anxiety disorders, and found that according to Beighton criteria, 52.7% of children with anxiety disorders had GJH.²² In contrast, 16% of children in the comparison group had GJH, and age was found to be a risk factor, whereas sex, severity and type of anxiety disorder and ADHD were not predictive of GJH. (p 5, lines 15-29)

4. You mentioned that after correcting for age and BMI, heart rate at 14 years was significantly associated with both GJH at age 14 years, and also with depression at 18 years (page 10, lines 33-36), elevated resting heart rate, mediated this effect. (Abstract, line 32). Please add more information about physiological changes in depression.

We have added more information in the introduction:

Differences in cardiovascular autonomic activity (e.g., heart rate) between individuals with and without psychiatric conditions have been observed, where higher resting heart rate predicts subsequent psychological symptoms in a large longitudinal cohort study.^{18 19} Symptoms of depression are associated with altered autonomic activity,^{20 21} such as increases in electrodermal activity,²² skin temperature and respiratory frequency,²³ and decreased heart rate variability.²⁴ (p 4, lines 54-60 & p 5, lines 3-9)